# The Revolution of Lateral Flow Assay in the Field of AMR Detection

**DOI:** 10.3390/diagnostics12071744

**Published:** 2022-07-19

**Authors:** Hervé Boutal, Christian Moguet, Lilas Pommiès, Stéphanie Simon, Thierry Naas, Hervé Volland

**Affiliations:** 1Département Médicaments et Technologies Pour la Santé (DMTS), Université Paris Saclay, CEA, INRAE, SPI, 91191 Gif-sur-Yvette, France; herve.boutal@cea.fr (H.B.); christian.moguet@cea.fr (C.M.); lilas.pommies@cea.fr (L.P.); stephanie.simon@cea.fr (S.S.); 2Bacteriology-Hygiene Unit, APHP, Hôpital Bicêtre, 94270 Le Kremlin-Bicêtre, France; thierry.naas@aphp.fr; 3Team Resist, UMR1184, Université Paris-Saclay—INSERM—CEA, LabEx Lermit, 91190 Gif-sur-Yvette, France; 4Associated French National Reference Center for Antibiotic Resistance: Carbapenemase-Producing Enterobacteriaceae, 94270 Le Kremlin-Bicêtre, France

**Keywords:** lateral flow immunoassay, antibiotic resistance, ESBLs, carbapenemases, detection

## Abstract

The global spread of antimicrobial resistant (AMR) bacteria represents a considerable public health concern, yet their detection and identification of their resistance mechanisms remain challenging. Optimal diagnostic tests should provide rapid results at low cost to enable implementation in any microbiology laboratory. Lateral flow assays (LFA) meet these requirements and have become essential tools to combat AMR. This review presents the versatility of LFA developed for the AMR detection field, with particular attention to those directly triggering β-lactamases, their performances, and specific limitations. It considers how LFA can be modified by detecting not only the enzyme, but also its β-lactamase activity for a broader clinical sensitivity. Moreover, although LFA allow a short time-to-result, they are generally only implemented after fastidious and time-consuming techniques. We present a sample processing device that shortens and simplifies the handling of clinical samples before the use of LFA. Finally, the capacity of LFA to detect amplified genetic determinants of AMR by isothermal PCR will be discussed. LFA are inexpensive, rapid, and efficient tools that are easy to implement in the routine workflow of laboratories as new first-line tests against AMR with bacterial colonies, and in the near future directly with biological media.

## 1. Introduction

Since their first use in the late 1930s, antibiotics have become a valuable tool in the fight against bacterial infections, leading to a considerable improvement in clinic and public health [1,2]. However, subsequent to their use, overuse, and misuse, bacteria resistant to these molecules have emerged [3,4,5]. Antimicrobial resistance (AMR) is today universally recognised as a global threat because of the rapid emergence and dissemination of resistant bacteria and genes among humans, animals, and the environment on a global scale. AMR thus represents a heavy burden for healthcare systems all over the world. ESKAPE pathogens (*Enterococcus faecium, Staphylococcus aureus, Klebsiella pneumoniae, Acinetobacter baumannii, Pseudomonas aeruginosa et Enterobacter* spp.) combined with antibiotic resistance have greatly increased the risk of morbidity and mortality, especially in ICU settings [6].

Moreover, resistant bacteria are easily transferred from one reservoir to another, and consequently so are the resistance genes they carry [7].

AMR hinders the fight against infectious diseases [8], and the development of new antibiotics is slower than the emergence and spread of resistant organisms [9]. Their dissemination has obvious health impacts, but also economic effects in high- as well as in low- and middle-income countries [6,10,11]. Indeed, actions have to be taken, and the One Heath approach aims to ensure that antimicrobials are optimally used in both human and animal health, but also in agriculture. In the veterinary field, the amounts of antibiotics used, especially those with an important medical value, have been significantly decreased in order to preserve their efficacy in human medicine [12]. Another essential action is the early detection of resistant bacteria in clinical settings to allow implementation of efficient infection control measures and identification of resistance mechanisms so that the most appropriate antibiotic therapies can be proposed. Implementation of rapid point of care (PoC) diagnostic tests is mandatory to achieve this goal [13].

In general, for clinical diagnostic laboratories to identify an antimicrobial-resistant pathogen, it must first be isolated from the matrix (blood, urine, rectal swab) before antimicrobial susceptibility testing (AST) can be performed. AST is a bacterial growth test in the presence of antibiotics that allows the pathogen’s susceptibility or resistance to a given antibiotic to be determined. Several techniques are available such as disc diffusion (e.g., Bio-Rad, Oxoid, Hercules, CA, USA), broth dilution (e.g., Micronaut [Bruker]; Sensititre [Thermo Fisher Scientific, Waltham, MA, USA], antibiotic gradient test (e.g., E-test [BioMérieux]), and also automated systems (e.g., Vitek systems [BioMérieux]; Phoenix [Becton Dickinson Diagnostic Systems]; Microscan WalkAway *plus* system [Beckman Coulter]). These tests are often time-consuming, and results generally take 16–24 h.

Nucleic acid amplification technology (NAAT) such as that used in PCR-based technics, both conventional and real-time formats, can be performed. These tests target genetic determinants of resistance. Compared to phenotypic resistance tests, they can be performed directly using clinical samples with a shorter turnaround time providing earlier information regarding the resistance profile of the clinical strain. Nevertheless, the presence of the resistance gene is not always correlated with a phenotypic resistance, which may also depend on its level of expression [14].

Matrix-assisted laser desorption/ionization time-of-flight mass spectrometry (MALDI-TOF MS) in the context of resistance profile determination can be used to highlight the hydrolysis of antibiotics while incubated with a strain by detecting degradation products [15].

All these methods are complex and require technical skills and expertise or depend on expensive equipment. In case of an infection with a resistant isolate, the situation requires immediate, on-site identification with rapid, economical, and user-friendly methods. In this regard, the lateral flow assay technique (LFA), also known as rapid diagnostic test (RDT), has proven valuable in the detection and identification of antibiotic resistant isolates. LFA meets all the prerequisites defined by the World Health Organization for an ideal PoC test, or in general for any point of need (PoN) test, known as ASSURED [16,17] (Affordable, Sensitive, Specific, User-friendly, Rapid and robust, Equipment-free and Deliverable). Moreover, LFA can integrate the P5 medicine model, which is becoming prominent in the healthcare system. This model relies on a personalized, predictive, preventive, and participatory approach, with as a final objective the implementation of therapy most suitable for the patient [18].

In this paper, we review the use of LFA for early and rapid identification of AMR strains. These assays are based on the interaction of antigen-antibody (Lateral Flow ImmunoAssay: LFIA) or DNA-DNA hybridization (Nucliec acid Lateral ImmunoAssay: NALFIA or Nucleic Acid Lateral Flow: NAFL) [19,20,21].

## 2. General Presentation of LFIA

### 2.1. Components and Principle

LFIA tests generally consist of a strip supporting different porous compounds on which liquids migrate by capillarity. The sample pad (SP), which is usually made of cellulose, receives the sample that successively drains onto the conjugate pad (CP). Made of glass fiber, the CP is the storage area for the conjugate, a labelled molecule that generates the signal on the strip. This signal is located on a nitrocellulose membrane printed with different compounds in order to form one test line and a control line (Figure 1). The test line captures the targets of interest while the control line serves as an inner control for confirmation of correct flow and compatible test conditions. Finally, the wicking pad acts as a pump as well as a reservoir for the liquid dispensed on the SP, and its capacity influences the volume of sample that can be analyzed. All components overlap one over the other and are generally enclosed in a plastic cassette. This cassette provides pressure points to maintain close contacts, providing efficient flow of the reagents through the strip, protection, easy handling, a localized sample dispensing area, and a reading frame.

Migration begins once the sample has been loaded on the SP that, depending on the nature of the samples, can be pre-treated in order to reduce matrix effects. The sample solution resuspends the conjugate that forms complexes with the analyte if present. Capillary pressure transports the complexes along the nitrocellulose membrane, and they accumulate on the test line and the excess of conjugate on the control line.

Most of the time, the test line and the conjugate involve antibodies specific for the target being detected, and two formats of LFIA are available. The first, known as non-competitive or sandwich immunoassay format, is for large molecular weight analytes such as proteins that provide several antigenic sites. In this case, a colored test line represents a positive result. The second format, known as competitive or inhibition immunoassay format, is dedicated to small molecular weight antigens, and an attenuated or absent test line indicates a positive result.

### 2.2. Advantages of LFIA

LFIA are one-step assays that require no washing and only a small sample volume. The time to results, following an easy sample handling, is short (15–30 min), which positions LFIA as a good candidate for primary screening at PoC or PoN. They are inexpensive and do not have specific storage constraints, such as refrigeration, making them accessible to third-world countries [22,23]. The qualitative interpretation of the result can be visual, and no particular skills are required. Moreover, depending on the parameters of the test, results can also be semi-quantitative or even quantitative through the use of a reader [24,25]. In addition, if the reagents are available, the development time to market of an LFIA is relatively short, which can be useful to respond to an urgent sanitary crisis, as illustrated with the recent COVID-19 pandemic. The LFIA, due to its configuration, allows the detection of multiple analytes such as proteins, haptens, or nucleic acids [20].

Interest in LFIAs is best illustrated by the increase in publications describing their use in different fields of application over the last 10 years, especially in the clinical domain (Figure 2).The search query used was (Scopus format): TITLE-ABS-KEY (lateral AND flow AND immunoassay) OR TITLE (lateral AND flow AND assay) OR TITLE (immunochromatographic) AND PUBYEAR > 2009 AND PUBYEAR < 2020 AND (LIMIT-TO (DOCTYPE, “ar”)) AND (LIMIT-TO (LANGUAGE, “English”)). The research has been done in October 2020. Then, the articles were selected following the PRISMA guideline for systematic reviews [22]. It is likely that the ease of use, speed, and specificity of these tests are at the origin of their exponential use, including in the field of AMR.

## 3. Classical Lateral Flow Assays and AMR

Bacteria possess four main mechanisms that can confer resistance to antibiotics: (i) expression or overexpression of efflux pumps reducing antibiotic concentration within the bacteria; (ii) decreased permeability of the membrane or cell wall leading to ineffective drug concentration in the bacteria; (iii) changed target structure with affinity loss for antibiotics; and (iv) antibiotic degradation mediated by hydrolysis [26]. Any protein involved in a mechanism of AMR can then become a privileged target for LFIA. As a result, a number of LFIA tests have been developed, and some are commercially available.

### 3.1. Monoplex LFIA to Address AMR Detection

As the number of antibiotic-resistant isolates has increased, LFIA tests have been developed to target enzyme-mediated resistance traits in the clinically most important bacteria, with the primary objective of detecting enzymes involved in the resistance of most prevalent pathogens. A LFIA test detecting the expression of the *Pseudomonas aeruginosa* 6′-N-acetyltransferase AAC(6′)-Iae, which confers resistance to aminoglycosides, was described in 2010 and has a sensitivity of 10^5^ cfu/test [27]. Another LFIA targets ArmA 16S rRNA methylase, which is one of the most prevalent 16S rRNA methylase reported and leads to pan-aminoglycoside resistance in Gram-negative bacteria (GNB) such as *Acinetobacter baumannii* and *Escherichia coli* [28].

Methicillin-resistant *Staphylococcus aureus* (MRSA) is another major pathogen responsible for severe morbidity and mortality in many hospitals worldwide, which also has an efficient capacity for spreading in the community [29]. The early detection of methicillin resistance, which confers resistance to all ß-lactams, is essential. It relies on the detection of the penicillin-binding-protein 2a (PBP2a), which has a reduced affinity for beta-lactam antibiotics. Antibody-based techniques for the detection of MRSA are challenging as *S. aureus protein A* binds to mammalian immunoglobulins. Nevertheless, the use of IgY anti-PBP2a antibodies has been described by Yamada et al. [30], and another study described an optimized LFIA with a detection limit of 10^4^ cfu/mL [31].

Vancomycin-resistant enterococcus (VRE) is one of the most important nosocomial pathogens worldwide [32,33,34]. The vancomycin resistance mechanism in *Enterococcus faecium* and *Enterococcus faecalis* is mostly acquired and linked to the production of ligases. In Europe, the most prevalent are VanA and VanB [32,35]. In this context, a LFIA for the identification of VanA-VRE isolates was described with 100% sensitivity and 100% specificity and a limit of detection of 6.3 × 10^6^ cfu and 4.9 × 10^5^ cfu per test when the growth was performed on MH or ChromID^®^ VRE plates, respectively [36]. Another LFIA for the detection of VanB-VRE isolates with a lower sensitivity and a mandatory pre-culture on vancomycin-containing media for induction of VanB ligase has also been reported [37].

Beta-lactams represent the major family of antibiotics to treat infections due to Gram-negative bacteria, but their use is currently challenged by the spread of beta-lactamases [38]. In particular, the spread of extended-spectrum beta-lactamases (ESBLs) among Enterobacterales represents a major threat as these enzymes are able to inactivate most beta-lactam molecules (including 3rd and 4th generation cephalosporins and aztreonam), sparing only carbapenems [39]. The most common family of the ESBL, the CTX-M, represented by five sub-groups: CTX-M-1, CTX-M-2, CTX-M-8, CTX-M-9, and CTX-M-25, has disseminated worldwide [40]. The increase in the prevalence of ESBL-producing Enterobacterales has led to an increased use of carbapenems, a last resort antibiotic, to treat infections with ESBL-producers. This has led to the selection and subsequent increase in bacteria resistant to these antibiotics. Carbapenem-resistant Enterobacterales (CREs) are usually resistant to most, if not all antibiotics, thus posing serious therapeutic issues in clinical practice. Among CREs, carbapenemase-producing isolates are the most worrisome, as they are capable of efficiently hydrolyzing carbapenems, and their genes are carried by plasmids that may be exchanged between bacteria. Early identification of carriers is essential in order to implement reinforced infection control measures, among which isolation of the patient is a prerequisite. There are five major carbapenemases, KPC, NDM, OXA-48, VIM, and IMP. LFIAs to detect KPC and OXA-48-like enzymes [41] (Coris Bioconcept, Gembloux Belgium), IMP (the most prevalent metallo-β-lactamases in Japan [42,43]), and NDM [44,45] have been developed, all with 100% sensitivity and specificity when used with isolated colony from agar plates.

Faced with infections resistant to carbapenems, the paucity of therapeutic options has led to the use of polymyxins such as colistin as last resort antibiotics [46]. Inexorably, bacteria acquired colistin resistance. While most colistin resistance is due to chromosomic mutations, plasmid-encoded mechanisms, such as MCR-1, have also been described. The latter are considered particularly threatening, as the mcr genes are plasmid encoded, and the resistance phenotype is difficult to detect. MCR-1, initially described in 2015, mediates the modification of the lipopolysaccharide by a phosphoethanolamine transferase activity [47]. More than eight different MCR-alleles have now been described [48]. Soon after the first description of MCR-1, a LFIA directed against this allele was commercialized (NG-Test MCR-1). This assay has been evaluated in a multicentric study against a collection of human and animal enterobacterial isolates. The results revealed 100% sensitivity for MCR-1 expressing isolates, but some MCR-2 carriers were missed, and the assay did not detect MCR-3, MCR-4, and MCR-5 carriers [49].

### 3.2. Multiplex LFIA in the AMR Field

Monoplex LFIAs have proven efficacy for the detection and identification of resistance-determining markers. When targeting mechanisms involving a multitude of variants, such as the CTX-M family, a wider specificity can be relevant. Multiplex LFIAs, able to detect CTX-M enzymes but without discriminating the variant or subgroup to which they belong, have been developed. For example, the NG-Test CTX-M MULTI (NG Biotech, Guipry, France) is commercially available and relies on a cocktail of anti-CTX-M mouse antibodies, immobilized on a unique test line (Figure 1, central panel). This test allows the detection of the five CTX-M-subgroups. A recent study showed that the NG-Test CTX-M MULTI could detect 98% of ESBL-producers from a French clinical setting, either from colonies or positive blood cultures, missing only two SHV-ESBL producers [50]. A study, conducted in Italy, reported that the NG-Test CTX-M MULTI was a reliable assay for the detection of CTX-M-like ESBLs from bacterial pellets from blood culture broth, showing excellent sensitivity and specificity [51]. A further study recently described the detection of CTX-M-group-1, -2, and -9 producers using a monoclonal rabbit anti-CTX-M antibody, and showed 100% sensitivity and specificity with clinical isolates grown on agar plates [52]. Most resistance genes are carried by mobile genetic elements [53] and a single strain can harbor more than one resistance determinant. In this scenario, the detection and identification of more than one mechanism of resistance in a single test is relevant. To do so, several test lines are printed on the same strip (Figure 1, lower panel) and target identification is made through a spatial repartition [54]. Several multiplex assays have been described, so far only for the identification of carbapenemase-producing strains. Several versions of the RESIST LFIA test exist (from Coris Bioconcept), which are differentiated by the number of carbapenemases that can be detected. RESIST-3 [55] can detect NDM, KPC, and OXA-48 enzymes. This test has since been upgraded with the additional detection of VIM (RESIST-4 [56]), and IMP or OXA-163 (RESIST-5 [57,58,59]). All these multiplex assays consist of two-independent cassettes that are used in parallel with the same bacterial extract. Another assay, named NG CARBA-5 [60] (from NG Biotech, Guipry, France), also targets NDM, IMP, VIM, OXA-48, and KPC carpabenemases, but a major difference compared to the RESIST-LFIA test is that the sample has to be loaded onto one unique cassette. The NG Carba-5 has been evaluated by the Antimicrobial Resistance and Healthcare Associated Infections (AMRHAI) in London, with isolates covering the diversity of the carbapenemases. It showed 97.31% sensitivity and 99.75% specificity, missing IMP-13- and IMP-14-like enzymes [61], also not identified during a previous evaluation by the same team [62]. Missing such enzymes could be an issue in countries with a high IMP prevalence, and since the AMRHAI study, a new version has become available with improved detection of IMP variants [63]. In Europe, NG CARBA-5 shows high sensitivity (97.3% to 100%) and specificity (96.1% to 100%) according to recent studies [61,64,65,66,67]. NG CARBA-5 has also received U.S. Food and Drug Administration clearance and evaluation at three medical centers in the USA confirmed its accuracy for detecting and identifying the five most common carbapenemases [68]. NG Carba-5 is often compared to the molecular Xpert Carba-R test (Cepheid) and shows a very high correlation with Carba-R [68,69,70,71], but with the advantages of time efficiency and lower cost. LFIAs targeting carbapenemases using the CIM method and its various versions (mCIM, zCIM, and eCIM) make detection of most common and rare variants rapid, simple, and inexpensive [72,73]. Most studies have described the use of NG Carba-5 with Enterobacterales isolates, but in France, it is also valuable for the detection of carbapenemases produced by *Pseudomonas* spp., 89.4% detected compared with only 12.9% of carbapenemase-producing *Acinetobacter* spp. Indeed, the most prevalent carbapenemases in this organism are not targeted by this test [64].

### 3.3. Limitations of LFIA in the Context of AMR

Despite the many possibilities that LFIA offer, there are some limitations. For example, variations in sample volume loaded on the device can both reduce the accuracy of the result and impact the sensitivity of the test. Moreover, LFIA performance relies mostly on antibody affinity and specificity. However, even if the latter is determinant, we can only detect what we are looking for; thus, the specificity is a limitation in respect of the diversity of enzymes involved in bacterial resistance mechanisms. Implementation of an LFIA test in the clinical setting has to take into account the local epidemiological context in terms of prevalence of resistance mechanisms. The user should keep in mind that any new enzyme variant harboring a mutation in the epitope recognized by any of the antibodies involved in the test may give a reduced signal or a false negative result. Moreover, isolates with an AMR profile but which harbor a mechanism not targeted by the assay will also give a negative result. As for all diagnostic tests, interpretation of results must be made in light of the clinical data and viewed critically. Most LFIAs have been evaluated on colonies grown on agar-containing plates. The ability to use directly from clinical samples would therefore be an important improvement as it would increase turn-around time. For this to be achieved, the sensitivity of the assays has to be improved (on average 10^5^–10^6^ cfu), and interferences with the different biological matrixes needs to be evaluated. The sample or the matrix analyzed may require an additional sample pre-treatment to avoid interferences. Indeed, sample viscosity may prevent efficient migration on the nitrocellulose membrane leading to invalid results, or the matrix may generate interferences leading to false positive or negative results. Current LFIA systems must therefore be improved before they can be used directly with clinical samples [20,74]. Finally, LFIAs only detect the enzymes for which they have been developed. This is the case for the NG-Test CTX-M MULTI that detects the main ESBLs and CTX-Ms, but misses minor ESBLs, and plasmid-encoded cephalosporinase. Combining it with an LFIA that detects hydrolytic activity instead of the enzyme itself would therefore be a major improvement.

## 4. LFIA: A Phenotypic Method

### 4.1. Classical Phenotypic Methods

Unlike LFIAs, phenotypic methods allow the detection of all enzymes and variants while maintaining good sensitivity and specificity. They detect the enzymatic activity using a panel of antibiotics, given as an indicator of resistance. These multiple methods are based on different technologies (each test is described in the Figure 3).

The diffusion disc test (antibiogram, combined disc synergy test, inhibition test), is one of the reference methods [75,76,77,78]. The principle consists of placing the culture of bacteria in the presence of one or more antibiotics and observing the consequences on their development and survival in a Petri dish (Figure 3A).The modified Hodge test consists of inoculating Mueller–Hinton agar with the reference bacterial strain, wild type Escherichia coli ATCC 25,922, at 0.5 McFarland diluted 1:10. A disc containing carbapenem (meropenem or ertapenem) is placed in the centre of the agar. Colonies of the test strain are picked and plated in a line from the disc to the periphery of the plate. The presence of a carbapenemase is demonstrated by a cloverleaf indentation of Escherichia coli ATCC 25,922 that develops along the growth line of the isolate in the diffusion zone of the disc [79] (Figure 3B).The colorimetric tests detect β-lactamase activity via a color change in the reactive medium related to hydrolytic activity. This change in color is due to a biochemical change in the medium such as acidification. Here, three of them are presented: the Carba NP test, the β-Carba test, and the β-Lacta test [80,81,82] (Figure 3C,D).The electrochemical test is based on the analysis of conductivity variations within an electrode, composed of eight probes, coated with a conductive polymer, polyaniline. This variation is induced by changes in pH and redox potential related to an enzymatic hydrolysis reaction of imipenem [83] (Figure 3E).Mass spectrometry is used to detect the degradation of antibiotics by measuring their mass. Currently, among the different mass spectrometry technologies the MALDIO-TOF is the most used for this application. After ionization, the ionic molecules are accelerated in an electric field and projected towards a detector. This detector allows the ions to be separated and analysed according to their time of flight, which depends on their mass [84,85] (Figure 3F).The selective media tests are composed of chromogenic substances, rich nutrients, as well as specific antibiotic depending on the desired detection. Thus, they allow the identification of the strains involved through differential staining induced by the presence of characteristic enzymes [86] (Figure 3G).The carbapenem inactivation method consists of lysing the bacterial colonies to be tested in order to recover the suspension containing the possible β-lactamases. This lysate is divided into two separate tubes containing, respectively, a 10 µg meropenem disc (carbapenemase detection) and a 5 µg cefotaxime disc (ESBL detection). In parallel, two Mueller–Hinton agar plates were inoculated with an Escherichia coli ATCC 25,922 strain, known as a β-lactam sensitive strain. After incubation for 2 h at 36 °C, the meropenem and cefotaxime discs are transferred successively to the two plates. A further incubation is carried out for 4 h at 37 °C, followed by a reading and interpretation of the diameter of the inhibition zones. The absence of inhibition zones around the disc reflects degradation of the antibiotic during the first incubation, indicating hydrolytic activity of β-lactamases [87,88,89] (Figure 3G).Ultraviolet (UV) spectroscopy is an electron spectroscopy technique allowing the detection of the hydrolysis of the β-lactam core by a difference in absorbance between the non-hydrolysed and hydrolysed form (called ΔA), at a given wavelength [90,91] (Figure 3I).

**Figure 3 diagnostics-12-01744-f003:**
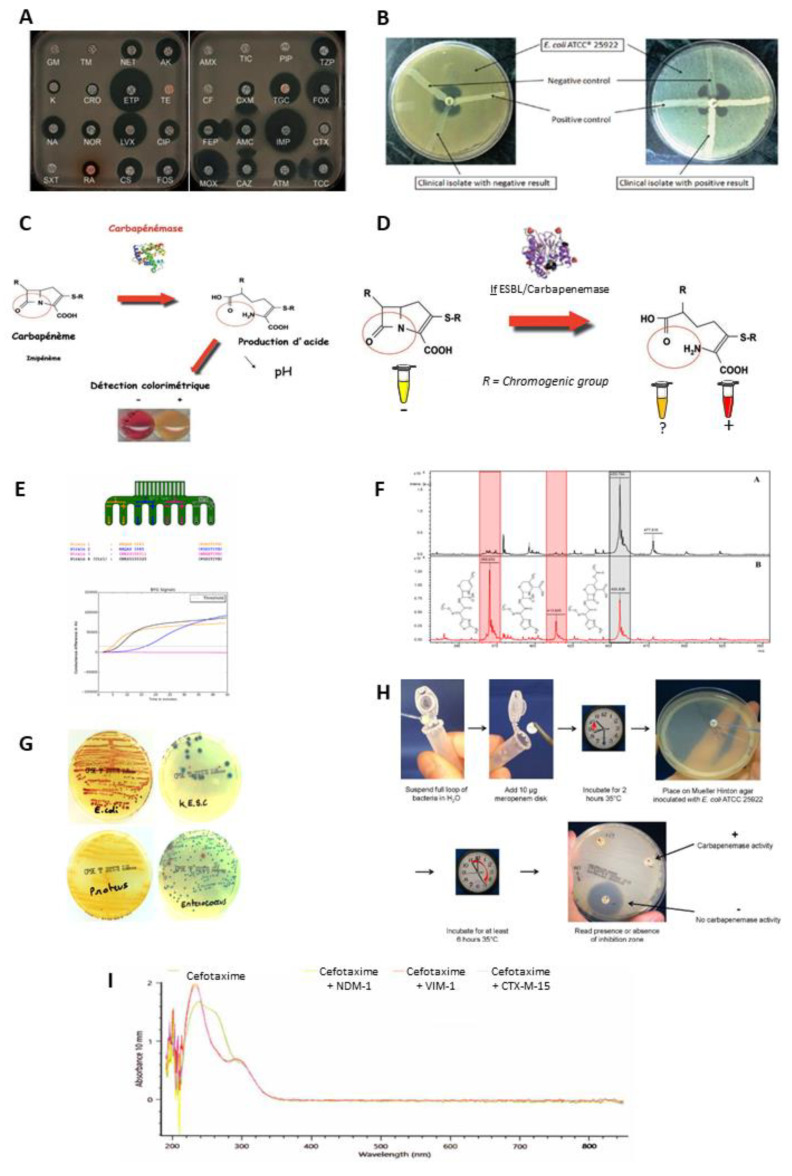
(**A**) The disc diffusion antibiogram is based on the use of several classes of antibiotics and measures the diameter of the zone of growth inhibition [78] (Reprinted/adapted with permission from [78]. 2014, Nordmann P. and Poirel L.). The smaller the zone, the stronger the resistance to the antibiotic. Other techniques such as the E-test or broth micro-dilution can be used to determine the minimum inhibitory concentration (MIC) to complete the data. Antibiotic susceptibility testing can often be accompanied by inhibition tests based on the synergy between β-lactamase inhibitors and β-lactams. Based on the inhibition zone diameters, it is possible to orient resistance towards a β-lactamase class. (**B**) The modified Hodge test, considered as a confirmatory test, is based on the ability of a resistant strain to hydrolyze a third-generation cephalosporin or carbapenem to allow the growth of a non-resistant *E. coli* strain, which is thus synonymous with bacterial resistance [92]. (**C**) The Carba NP test or the ESBL NDP test uses a colored indicator to track a change in pH. The hydrolysis of a β-lactam induces the opening of the β-lactam ring and an acidification of the medium [78] Reprinted/adapted with permission from [78]. 2014, Nordmann P. and Poirel L.) (**D**) The β-lacta or β-Carba are based on the use of chromogenic substrates. The hydrolysis of the latter induces the release of the chromophore, which causes a change in the color of the medium. (**E**) The electrochemical test is based on the use of electrodes that register differences in conductance when the β-lactam substrate is hydrolyzed [93]. (**F**) Mass spectrometry highlights the structural changes that a β-lactam undergoes upon hydrolysis. Thus, the appearance of peaks at expected molecular weights indicates the presence of bacterial resistance [94]. (**G**) Chromogen-based selective media are composed of specific antibiotics and a chromogen substrate. The hydrolysis of the substrate and the antibiotics present are representative of a type of resistance. Thus, there are several selective media to identify the resistance group [95]. (**H**) The CIM is based on an antibiotic disc that is incubated for 2 h with a resistant strain. This is collected and placed on an agar plate containing non-resistant *E. coli*. The diameter of the inhibition zone thus reflects resistance [96]. (**I**) The UV spectrophotometric technique involves measuring the spectrum of a substrate before and after incubation with a resistant strain. The opening of the β-lactam cycle is reflected by a decrease in absorbance at wavelengths around 260 nm.

Some methods are associated with high costs, require dedicated equipment, long training courses (MALDI-TOF, UV spectrophotometry), or require that kits be commercialized (BYG Carba). Other methods, such as LFIA, are often inexpensive, have rapid results, require low staff skill levels, and are easy to use (disc diffusion antibiogram, modified Hodge test, Carba NP, β-lacta or β-Carba, Chromogen-based selective media, CIM). However, these methods also have some limitations such as: difficulty differentiating enzymes within a resistance group such as extended-spectrum β-lactamases, carbapenemases, and plasmid-mediated cephalosporinases [77]; sensitivity and specificity of the tests dependent on β-lactamases and their hydrolytic activities [97]; and sometimes ambiguous interpretation, especially for colorimetric tests, which may leave room for misinterpretation [98]. It is therefore important that laboratories are vigilant in the interpretation and identification of certain emerging resistances. In some cases, confirmation by genotypic methods is necessary, delaying the delivery of results.

The following table summarizes the characteristics of different tests (Table 1). For the commercialized tests, some suppliers have been stipulated.

It may therefore be interesting to combine the advantages of phenotypic tests (disc diffusion antibiogram, modified Hodge test, Carba NP, β-lacta or β-Carba, Chromogen-based selective media, CIM) and LFIA tests in a new format.

### 4.2. Evolution of LFIA in the Detection of Antibiotic Resistance

#### 4.2.1. Detection of Enzymatic Activity Using LFIA

Recently, a LFIA was developed to monitor the hydrolysis of a cephalosporin, using an anti-cefotaxime monoclonal antibody. It is based on the very high specificity of the monoclonal antibody for the intact form of the cefotaxime, which is not recognized by this antibody after hydrolysis (Figure 4A(AB)). In this test format, the colloidal gold-labelled antibody recognizing intact cefotaxime is dried between the sample pad and the nitrocellulose membrane. On the nitrocellulose membrane, there is a test line (intact cefotaxime coupled to BSA) and a control line (antibodies recognizing the colloidal gold labelled antibodies). At the top of the strip, the absorption pad allows the sample to migrate along the strip (Figure 4B).

To conduct the test, the sample is grown overnight at 37 °C on culture medium. A single, isolated colony is selected and added to a buffer containing cefotaxime. Following an incubation period of 30 min, the solution is deposited on the test strip. After 10 min of migration, the results are read with the naked eye. There are two scenarios depending on whether or not the strain produces an enzyme with cefotaximase activity [99] (Figure 4C).

Unlike phenotypic assays, which generally detect hydrolyzed products, in this new test, the antibodies detect the intact antibiotic. Thus, the presence of enzymatic activity is reflected by the appearance of a signal on the test line, although this is a competitive test format. It allows an analysis similar to that of enzyme detection (positive test when a test line appears) and is simpler because it is easier to detect the appearance than the decrease of a signal. A study comparing this method to other phenotypic methods has been conducted and showed better performance and interpretation of results with LFIA [100].

The new test can detect the hydrolysis of third generation cephalosporins, regardless of the bacterial isolate or β-lactamase variant involved. It is based on the appearance of a signal that is easier to interpret than a slight color change used in colorimetric detection tests. The test is sensitive, specific, inexpensive and, produces rapid results. However, one of the disadvantages is that it does not, on its own, allow the identification of the enzyme involved, whether it is an ESBL, a carbapenemase, or an AmpC [101].

#### 4.2.2. Enzymatic Activity Detection Coupled to Enzyme Detection Using LFIA

Recently, we combined on one strip the enzymatic activity test and the previously developed CTX-Ms detection test [50]. This new format allows the simultaneous detection of the presence of β-lactamase activity and the identification of CTX-M enzymes with a specificity and sensitivity close to 100% [101].

These results open new perspectives for developing LFIA to detect other AMR mechanisms and so make efficient diagnostic tools available to most healthcare professionals.

At the same time, direct detection methods from clinical samples are of increasing interest because results would be obtained faster than from bacterial colonies. Nevertheless, this detection is often complex because the constituents of the media can interfere with LFIA.

## 5. Direct Detection of Antibiotic-Resistant Bacteria in Clinical Samples

The detection of antibiotic-resistant bacteria is performed with clinical samples such as blood cultures, urine, and rectal swabs. Between the collection of the sample and the rendering of the results, there is a very important step: the processing of the sample. This step modifies the characteristics of the sample so that it can be analyzed accurately by the selected technology. Depending on requirements, this may involve a varying number of steps, using different technologies, and having a variable duration.

The oldest (more than 100 years) and most commonly used method is the culture of samples on an agar plate [102]. This allows the concentration of bacteria to be increased and the eventual isolation of bacteria present in the sample. Depending on the composition of the media used, this method can allow the identification of bacterial strains [103]. The addition of antibiotic to the culture medium allows the direct detection of resistant bacteria. Sample culture can also be combined with the different methodologies used for the detection and identification of antibiotic resistance (molecular, immunological, biochemical, and mass spectrometry methods). Sample culture is inexpensive and simple but takes a long time (between 16 and 24 h).

Another processing method combines centrifugation and washing steps. It allows the bacteria from the clinical samples to be concentrated and the elimination of potential interferences due to elements present in the sample. This method has been applied to sample processing before AMR detection using LFIA [66], mass spectrometry [104,105], and biochemical techniques [106]. Although it may appear simple, a number of steps are involved, requiring pipetting and the use of a centrifuge. This process takes between 30 and 60 min.

Following the introduction of microfluidic and its application in the biological domain, new devices have been developed that integrate sample processing and target detection. All the steps in these assays, which use PCR or RT-PCR technology [107,108,109], are automated. These all-in-one tests have been applied to the direct detection of antibiotic resistance genes in clinical samples [110,111]. While these systems are easy to use and provide rapid results (around 1 h), they are expensive and require costly specific equipment which limits their use.

Until recently, immunochromatographic tests were used after sample culture on agar [60] or after centrifugation and washing steps [112].

### 5.1. Direct Detection Using Centrifugation/Washing Steps

Several studies [55,65,67,113,114] have shown that it is possible to detect carbapenemases directly from blood cultures using immunochromatographic tests (LFIA tests). In these studies, centrifugation and washing steps were performed to eliminate potential interferences present in the sample. The test was then performed using the final pellet and an extraction buffer. This procedure has been also used for the detection of CTX-M in blood culture [115]. Concentrations of bacteria in blood cultures are high and one study has shown that it would be possible to directly detect carbapenemases in blood cultures without centrifugation and washing steps [66].

Other teams have used centrifugation and washing steps for the detection of carbapenemases from rectal swabs using LFIA. However, for this sample matrix, prior incubation in the presence of an antibiotic is necessary [116,117].

### 5.2. Direct Detection Using a Dedicated Device

The objective of one of our research projects was to directly detect beta-lactamases in clinical samples using immunochromatographic tests in order to decrease the time to result by skipping the culture step. To maintain the advantages of LFIA tests, the sample processing had to be simple, economical, fast, without instrumentation, and had to integrate detection. To reach these objectives, we designed and produced using 3D printing a device named SPID for «Sampling, Processing, Incubation, Detection».

SPID has three elements: (i) a filtration element including a syringe adaptor, a cup with a membrane of 0.2 µm pore size, and a lower element; (ii) an extraction element including a cap with a plunger and a tank; and (iii) a detection element consisting of a plastic cassette integrating a lateral flow immunochromatographic strip. The filtration and extraction elements form the processing part (Figure 5).

For AMR detection with the SPID platform, the sample is collected with some air using a syringe. The syringe is then screwed on to the filtration device and the plunger pressed to filter the sample through the cup membrane. The filtration element is then opened, and the cup is transferred into the tank by sliding it inside. Extraction buffer is added to the cup and the tank is closed by screwing the cap. During this operation, the plunger of the cap will push the extract through the membrane into the tank. The tank is then placed on top of the cassette and pressed firmly until the operculum breaks down. This will release the extracted sample onto the strip, launching the migration. After 15 to 30 min, the results can be read with the naked eye (Figure 6).

The entire sample preparation takes about 2 min and the time to result with SPID for direct AMR detection in clinical samples is around 30 min.

This device has been evaluated in three different hospitals for the detection of CTX-M and the five major carbapenemases (NDM; KPC; OXA48; VIM; and IMP) directly in clinical samples. Three types of samples were used: blood culture, urine, and rectal swabs. The results showed that the device allows direct detection of these beta-lactamases in 30 min from blood cultures and urine, with a sensitivity and specificity between 98 and 100% for the detection of CTXM and between 94 and 100% for the detection of carbapenemases. For rectal swabs, an enrichment step is necessary to achieve the same performance for the detection of CTXM and carbapenemases (publication submitted).

This device is simple to use, low cost, and requires no equipment, and has the advantage of being adaptable to all existing immunochromatographic tests. These characteristics make the device perfectly adapted to the detection of AMR in the veterinary and environmental area.

## 6. Detection of Amplicons by LFIA

Conventionally, microbiology laboratories use phenotypic testing which involves incubating bacteria with antibiotics for AMR detection (disk diffusion, other systems calibrated by EUCAST). These methods take several days and require trained personnel. Rapid diagnosis helps reduce the spread of AMR, allows early isolation of the patient, and hence rapid and correct treatment. It also avoids unnecessary isolation and saves resources and money. Nucleic acid amplification technology (NAAT), immunochromatographic tests, electrochemical methods, microarrays, micro/nanoparticles, and mass spectrometry are all considered rapid detection tools. In this section, we will focus on nucleic acid amplification. With this powerful tool, it is not necessary to purify the nucleic acids, and the technology allows rapid diagnosis partly because it uses patient samples without enrichment and because many genes can be targeted from the same sample [118]. Furthermore, genetic-based tests are more accurate for tracing the spread of resistance genes [119].

### 6.1. Polymerase Chain Reaction (PCR)

Polymerase chain reaction (PCR) allows the replication and amplification of a target DNA through the Taq polymerase (or another thermostable enzyme). PCR is quite a fast test, sensitive, and reproducible [120]. It takes places in three steps: denaturation (95 °C), annealation (50–55 °C), and elongation (72 °C). These steps are repeated over several cycles to generate a large number of copies (30–35 cycles). Samples are then loaded on to agarose gel for electrophoresis and a single band for each target can be observed [121]. Multiplex PCR methods exist and can also be visualized by agarose gel electrophoresis [122]. There are two approaches: the first is known as specific PCR with specific primer targeting a single microorganism and the second as broad-spectrum PCR with primers targeting a gene (which may be present in several microorganisms) [120].

Many groups have developed different commercial kits depending on the application or the target (e.g., ThermoFisher, Biorad, Niotron, Sentinel, Norgen biotek corp, Biolabs, Altona diagnostic, Biobase, Redcaptain, and Biotron).

Unfortunately, PCR remains expensive for routine analytics or PoC testing as it is necessary to buy kits and complex devices for temperature variation.

Moreover, in complex matrices, inhibitors can disrupt amplification. This loss of information is avoided by processing the samples before analysis [119]. One of the possible solutions is to detect amplicons on LFA test. Some studies have used LFA as a detection method after PCR [123,124,125,126]. This technology is known as nucleic acid lateral flow immunoassay (NALFIA) or nucleic acid lateral flow (NALF). NALFIA involves reporter oligonucleotide probes, whereas NALF involves nucleic acid with hapten labels [74]. As PCR requires an instrument, the usefulness of a LFIA after PCR seems limited. Moreover, with real time PCR, detection is performed directly on the thermocycler.

Convective PCR (cPCR) is a new alternative to classic PCR that allows rapid amplification of DNA (less than 30 min) without complex equipment (Figure 7). The cPCR is performed on a heat block at 95 °C for 30 min, and the amplicons are analyzed on LFIA. With this technique, amplification and detection could be achieved with portable devices [127].

To use the PCR technique in PoC testing, several methods of amplification exist using particular enzymes that allow an amplification reaction at a constant temperature (NASBA [128], RCA [129], SDA [130], RPA [131], HAD [132], and LAMP [133]).

### 6.2. Loop-Mediated Isothermal Amplification (LAMP)

Loop-mediated isothermal amplification (LAMP) is an isothermal DNA amplification method developed by Notomi et al. in 2000 [133]. It can be performed in less than 1 h. The DNA polymerase (Bst polymerase) has a strong strand displacement activity, which allows the technique to be performed at a lower and constant temperature (between 60 and 65 °C). This method employs a set of specific primers that recognize different sequences on the target: the inner primer (FIP, BIP), the outer primer (F3, B3), and the loop primer (LF, LB). An illustration of the amplification process is provided in Figure 8 [134].

Compared to PCR, LAMP is more robust (DNA extraction not always required), less sensitive to inhibitor, and uses a smaller sample volume (blood, urine, stool). As a result, LAMP is associated with reduced costs and can be used in low-resource field settings [118].

LAMP has been applied to the detection of many pathogens (virus, bacteria, and parasites). In the last few years, more and more LAMP tests have been developed for the detection of antibiotic-resistant pathogens. The method is increasingly used in this field as it does not require complex equipment. Indeed, the amplification reaction can be realized using a heat block or water bath. The detection of amplification product can be analyzed using a range of common methods [136].

#### 6.2.1. *LAMP Detection Methods*

Several detection methods are compatible with LAMP (Figure 9). Indeed, the amplicons can be visualized by electrophoresis [137] and by monitoring the turbidity or the fluorescence. It is possible to look for color changes induced by chemical reaction [137,138,139]. The amplicons can be also detected using labelled primers with fluorescent dyes [140] or fluorescent intercalating dyes [137].

#### 6.2.2. LAMP Coupled to LFA Detection

Recently, LFA has been used as a tool for the detection of LAMP product. Indeed, few studies have focused on the use of LFA for the detection of antibiotic resistance genes [136,141,142].

To detect amplicons on a test strip, primers must first be labelled, e.g., with FITC, Texas red, biotin, Digoxygenin, FAM, or Hex. In addition to the labelled primers, other labelled probes or biotin-labelled dUTPs can be introduced into the reaction mix [136,143,144,145,146,147].

Some of the labeled primers involved in amplicon detection will be recognized by antibodies immobilized on the membrane. Frequently, streptavidin-gold-nanoparticles are dried and immobilized on the conjugate pad to reveal the captured amplicon [148] (Figure 10).

Gong et al. developed a LAMP assay combined with LFA for the detection of colistin resistance *mcr-1* gene [142]. The detection of amplicons is performed with a gold-nanoparticle-based lateral flow biosensor. The time-to-result (including sample treatment, LAMP reaction, and detection) is about 85 min, which is at least 90 min faster than *mcr-1* PCR. Moreover, the LAMP test is ten-fold more sensitive than the *mcr-1* PCR assay [142].

Multiplexing is important in the detection of antibiotic resistance as with a single test, several targets may be detected allowing the diagnosis to be refined. Chen et al. tried to simultaneously detect *mecA, femB*, and *nuc* genes. They compared multiplex PCR and three reactions with LAMP (one for each gene). The authors chose to detect LAMP amplicons with 2% agarose gel electrophoresis, and the addition of dye, which allows a color change of the solution. They found that multiplexing with these detection methods was not possible for LAMP. Indeed, a smear was observed on gel electrophoresis, and it was impossible to define which gene was amplified [149].

To solve this problem, Chen et al. developed a protocol to detect MSSA (targeting *femA*) and MRSA (targeting *mecA*) using LFA [141]. The FIP primers were labeled with digoxigenin (for *femA*) and with FAM (for *mecA*); both LF primers were also labeled with biotin. This protocol has been validated with biological samples and the results were similar using LFA-LAMP and traditional techniques. Similarly, Liu et al. developed a multiplex LAMP assay coupled with LFA to detect Mycobacterium abscessus and Mycobacterium massiliense. These mycobacteria are resistant to several antimicrobial agents [136].

The main drawback of LFA is that it is not a quantitative method. LAMP reactions are performed with small volumes that can limit the sensitivity of the test.

#### 6.2.3. Other Amplification Methods for AMR Detection

There are many isothermal amplification methods; those used for AMR detection are those that require the fewest steps and/or the least amount of enzyme. The methods used are as follows: GEAR, RPA, SDA, SMAP/SMAP2, HDA, NASBA, RCA, and TMA. [150,151]

LAMP remains the most suitable technique with a more efficient direct amplification, more resistance to matrix interference, and more sensitivity. In addition, LAMP seems to be more adapted to a POC application [150,151].

## 7. Conclusions

The development of LFA in the field of antibiotic resistance has continued to grow. The first generation of tests, based on the detection of the enzyme responsible for the resistance, avoided some of the limitations encountered in genotypic methods (detection of resistance genes by PCR) or in phenotypic methods (detection of enzymatic activity by modification of the antibiotic). The two main drawbacks of this first generation of tests were the non-detection of certain new variants of enzymes involved in resistance and the difficulty of detecting resistance directly in biological media. These issues have been overcome with the second generation LFA, which have consolidated their position as rapid, efficient, inexpensive, and easy-to-use tools. Additional developments include LFIA tests capable of directly detecting enzymatic activity (LFIA-CTX test), a SPID system to allow rapid and reliable detection in biological media with LFIA, and a LFA test associated with LAMP for improved test reliability and performance. These advances have considerably increased the involvement of LFA in the field of antibiotic resistance. All of the LFA tools are inexpensive, rapid, efficient, and are easy to implement in the routine workflow of laboratories as new first-line tests on bacterial colonies, and in the near future on biological media.

## Figures and Tables

**Figure 1 diagnostics-12-01744-f001:**
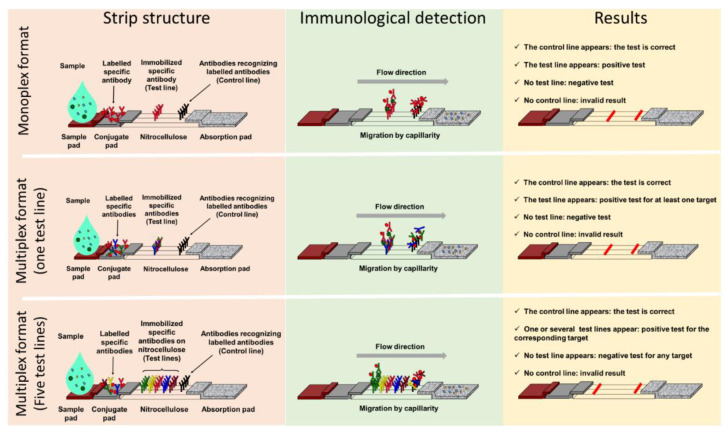
Lateral flow assay formats: components and principle. Presentation of three lateral flow immunoassay formats with, from left to right, their structure (pink panel), immunological detection principle (green panel), and results interpretation (yellow panel). The monoplex format able to detect only one target is presented in the upper panel, a multiplex detection format with one test line and no possible identification of the target is in the central panel, and a multiplex detection format with spatial separation of the test lines and identification of the target(s) detected is presented in the lower panel.

**Figure 2 diagnostics-12-01744-f002:**
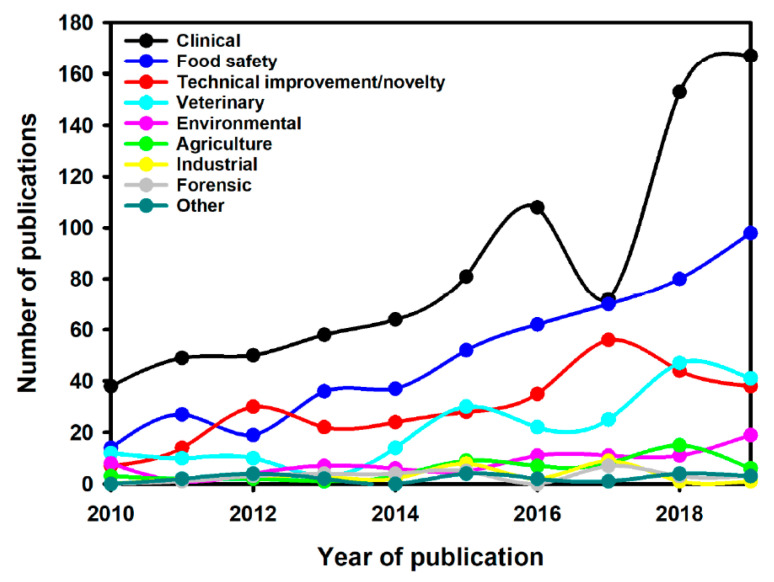
Number of publications per year for different LFIA applications [22].

**Figure 4 diagnostics-12-01744-f004:**
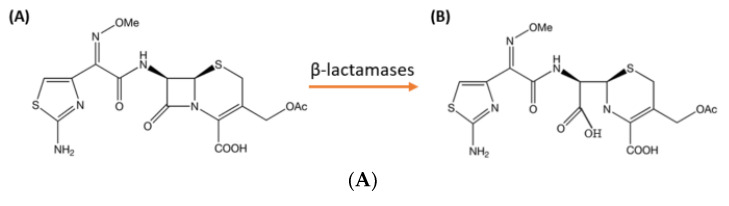
(**A**)**:** (A) Cefotaxime with β-lactam ring. (B) Hydrolysed cefotaxime with the open β-lactam ring. (**B**): Next to the sample pad, the colloidal gold-labelled antibody recognizing intact cefotaxime is dried. On the nitrocellulose membrane, there is a test line (intact cefotaxime coupled to BSA) and a control line (antibodies recognizing the colloidal gold labelled antibodies). The absorption pad allows the sample to migrate along the strip. (**C**) Case 1: In the absence of cephalosporinase expressing strains, the cefotaxime in the sample reacts with the anti-cefotaxime mAb after its re-solubilization. As all the mAb paratopes are occupied, the mAb cannot react with the cefotaxime immobilized on the test line. The mAb is immobilized by goat antibodies on the control line: if only one line appears, the test is negative. Case 2: In the presence of enzymatic activity, hydrolyzed cefotaxime is not recognized by mAbs, thus free paratopes are able to react with immobilized cefotaxime on the test line. A signal appears on the test line and the control line: if two lines appear, the test is positive.

**Figure 5 diagnostics-12-01744-f005:**
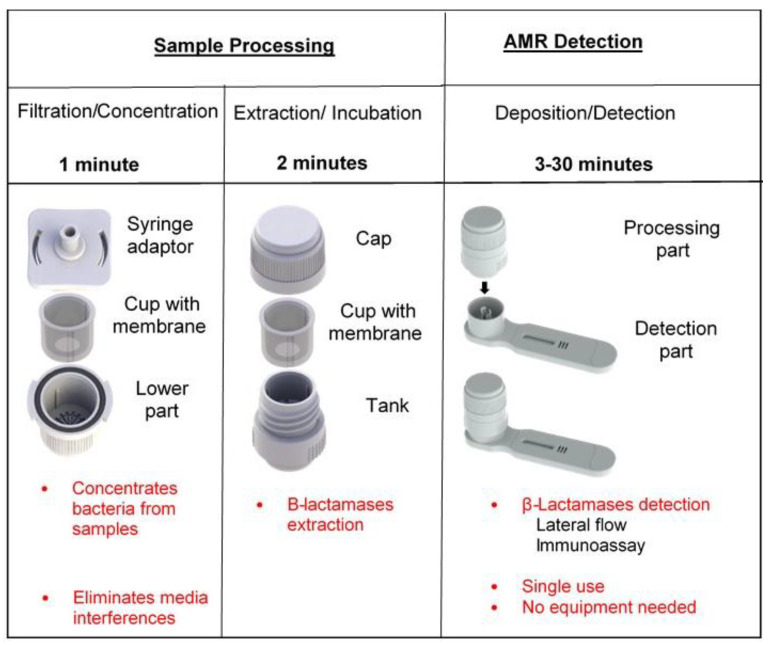
SPID platform elements.

**Figure 6 diagnostics-12-01744-f006:**
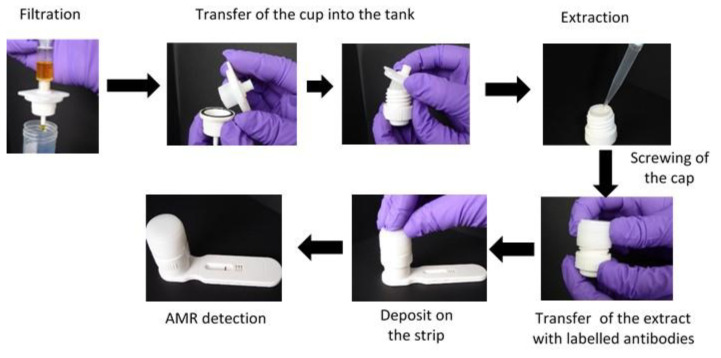
AMR detection workflow using the SPID platform.

**Figure 7 diagnostics-12-01744-f007:**
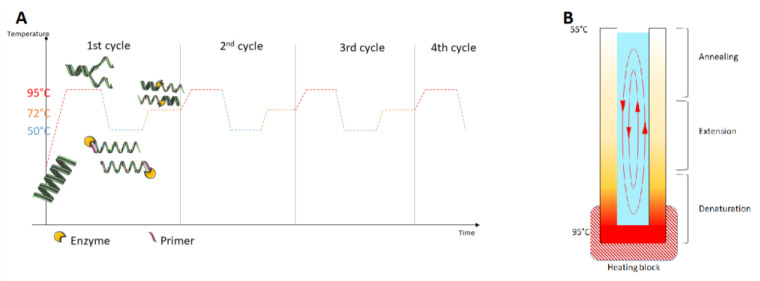
(**A**) PCR steps; (**B**) Principles of a cPCR.

**Figure 8 diagnostics-12-01744-f008:**
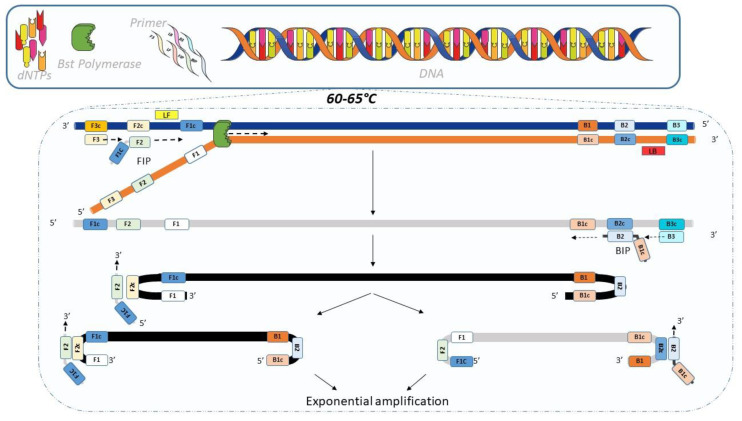
LAMP principle. The amplification starts when the FIP primer hybridizes to the F2c region of the strand. The F3 primer hybridizes to the F3c fragment on the DNA strand which initiates DNA synthesis by strand displacement. The strand bound to the FIP primer is then released and forms a loop structure at one end. This last structure allows the BIP primer to initiate subsequent DNA synthesis by strand displacement. This releases the sequence bound to BIP. The strand forms an altar-like structure and serves as the base for the LAMP cycles. The exponential amplification process then starts. BIP (or FIP) primer hybridizes to the loop structure. LF and LB primers are used to accelerate the amplification [135].

**Figure 9 diagnostics-12-01744-f009:**
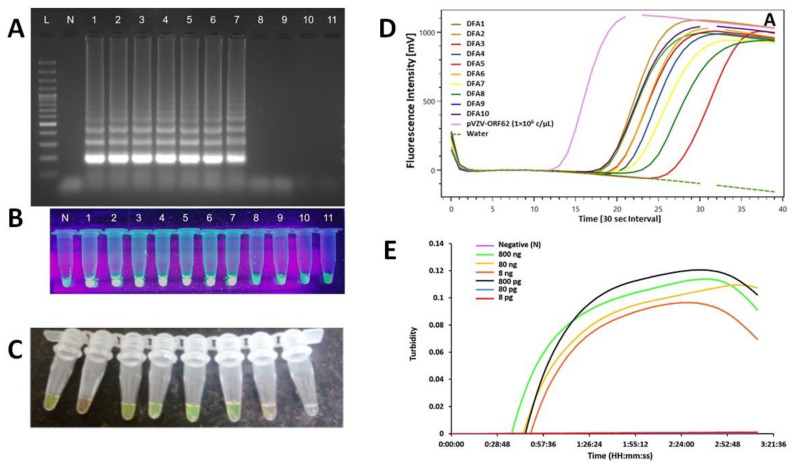
Detection methods for amplicons; (**A**) Gel electrophoresis [137]; (**B**) Chemical reaction, visual inspection with fluorescence [137]; (**C**) Chemical reaction, visual inspection with color change [139]; (**D**) Fluorescent primers, fluorescence monitoring [140]; (**E**) Chemical reaction, in real-time, turbidity monitoring [138].

**Figure 10 diagnostics-12-01744-f010:**
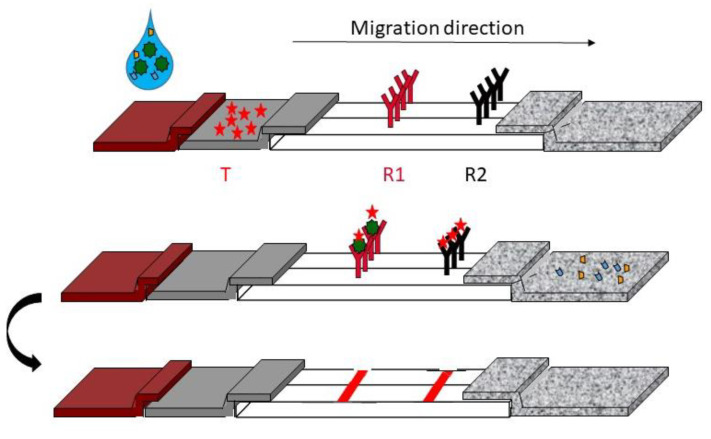
Example of lateral flow assay format for LAMP; R1: receptor 1, R2: Receptor 2, T: Tracker.

**Table 1 diagnostics-12-01744-t001:** A summary of the turnaround time, intrinsic performance, information provided, simplicity of performance, and major limitation(s) of each β-lactamase test.

Tests	Products	Time	Intrinsic Performance (Variable between Studies)	Information Provided	Easy to Implement	Main Limitation(s)
*Hydrolysis of β-lactams*	Antibiotic susceptibility test (from bioMérieux, Servilab…)	16–24 h	High	Points to ESBL or carbapenemases	Little expertise needed	Accumulation of large amount of information
	Chromogenic media (from bioMérieux, Oxoïd, CHROMagar)	16–24 h	Medium to high	Points to ESBL or carbapenemases	Little expertise needed	Non-detection of enzymes with low activity
	Modified Hodge Test *	16–24 h	Medium to high	Points to ESBL or carbapenemases	Medium expertise needed	Recurrent FP appearance + Difficulty in detecting MBLs
	Carba NP (from bioMérieux, Rosco Diagnostics)	2 h	Medium to high	Points to carbapenemases	Medium expertise needed	No standardisation Faulty interpretation (FN possible)
	Blue Carba (from Rosco Diagnostics)	2 h	High	Points to carbapenemases	Medium expertise needed	No standardisation Distorted interpretation (FN possible)
	β-Carba (from Bio-Rad)	0.5 h	High	Points to carbapenemases	Little expertise needed	Incubation >0.5 h for strains with 0XA-48 enzymes
	ESBL NDP (from Rosco Diagnostics)	2 h	High	Points to ESBL	Medium expertise needed	No standardisation Faulty interpretation (FN possible)
	β-Lacta (from Bio-Rad)	0.25 h	High	Points to ESBL	Little expertise needed	Overexpression of AmpC can lead to → FP
	Mass spectrometry (Bruker, BioMérieux, Beckman Coulter)	0.5–3 h	High	Points to ESBL or carbapenemases	Equipment needed + significant expertise needed	Visualisation of degradation products sometimes problematic
	UV spectroscopy *	1 h	High	Points to ESBL or carbapenemases	Equipment needed + significant expertise needed	Interference present + standardisation of OD difficult
	BYG Carba *	0.5 h	High	Points to carbapenemases	Equipment needed + significant expertise needed	No commercialised kit, nor evaluation of the technique
	Carbapenem inactivation method *	6 h	Medium to high	Points to ESBL or carbapenemases	Little expertise needed	Variability of inhibition zones → FN
*Inhibition of β-lactamases*	Combined discs (from Rosco Diagnostics, Mast Groups)	16–24 h	High	Points to ESBL or carbapenemases	Little expertise needed	Variable antibiotic distribution + some inhibitors not very effective
	Double synergy (from bioMérieux, Servilab…)	16–24 h	High	Points to ESBL or carbapenemases	Little expertise needed	Questionable zones of interpretation + some inhibitors not very effective
	E-Test (from Rosco Diagnostics, bioMérieux)	16–24 h	High	Points to ESBL or carbapenemases	Little expertise needed	Difficulty in detecting OXA-48 type enzymes + some inhibitors not very effective
*Lateral flow iiimmunoassays*	RESIST (from Coris)	0.25 h	High	Four of the five major carbapenemases in two tests (VIM, OXA-48, NDM, KPC)	Little expertise needed	Does not detect IMP enzymes or all new variants + use of 2 tests
	NG-Test (from NG-Biotech)	0.25 h	High	All five carbapenemases in one test + CTX-M	Little expertise needed	Does not detect all new variants

High intrinsic performance: >90%; medium: 70–90%; low: <70%; * No commercial tests.

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
