# Peer review of "The Revolution of Lateral Flow Assay in the Field of AMR Detection"

_diagnostics, 2022, doi:10.3390/diagnostics12071744_

Round 1

Reviewer 1 Report

Detection of antimicrobial resistant bacteria is very important, but difficult in realization task. Common microbiological techniques used for this purpose are very time- and labor consuming. However, recent developments propose significant improvements in this direction, and the use of lateral flow immune tests is one of the most perspective approaches for this purpose. Therefore, the review of Hervé Boutal and co-authors is actual and will be potentially interesting for wide row of readers. At whole, the prepared review provides a detailed and justified description of the current state of the art. However, a number of revisions seem to be necessary:

1. It is undesirable to use abbreviations in the title. Abbreviations LFIA and AMR are not of known knowledge. They make difficult quick and easy identification of the subject of the article/

2. Lines 57-60.  Please change «several days» to some specific interval. It would be reasonable also to extend the presented examples of available techniques considering products of other manufacturers (not only BioMerieux).

3. Stating the reasonable interest to LFIA in the Introduction, it would be useful to clarify the detected analytes that can be biomarkers of antibiotic resistance.

4. The authors reasonably indicate short time (about 15 min) that is necessary to perform LFIA. However, the assays confirming antibiotic resistance are often connected with prolonged preliminary actions with biosamples. So total time of the testing is an important parameter that should be considered in the review – as additional comments to specific examples or as an integrated table.

5. The section 2.1 repeats general information about the LFIA tests. It is really well-written and will be useful to readers, but the addition of some references to basic publications presented the action of LFIA tests with more details would be useful there.

6. Please take into consideration that the LFIA is the assay (procedure). It is not a device in contrast to LFIA test. So the statements such as «LFIA generally consist(S) of a strip » (line 88) are not correct. Please check the use of LFIA term throughout the manuscript.

7. Line 132. Lateral flow tests are successfully used for the detection of nucleic acids, but their recognition in the course of the assays is not immune reaction. So such tests do not realize namely LFIA.

8. Fig. 2. Please specify the criteria of search that were used in [19] to obtain data for this Figure.

9. Lines 140-145. A target for LFIA could be some compound, but not a mechanism. The cited list from [24] is useful, but it does not clarify targets. Some more specific justification concerning the targets would be useful here.

10. In the section 3 it would be reasonable to integrate comments about controlled biomarkers of antibiotic resistance as a separate sub-section before the consideration of monoplex and multiplex tests. Now the sub-sections 3.1 and 3.2 are overloaded by data about metabolic basis of the antibiotic resistance that ate useful but do not clarify construction and action of monoplex and multiplex tests.

11. The diagnostic sensitivity and specificity of the considered assays depend significantly from the studied populations. So it would be better to give some overall comments about these parameters, but exclude demonstration of specific details of different local applications.

12. Figure 3 contains some illustrations related to different phenotypic methods, but the figure is not discussed and thus does not help in understanding this variety. I think that a table listing principles of different techniques and their advantages would be more useful here. Such structure will provide better comments comparing to the existing non-structured list at lines 332-334.

13. Section 4.2.1. Homogeneous colorimetric assays are the most common ways in the detection of enzymatic activity. Are they used to detect the antibiotic resistance? What are the advantages of lateral flow tests for this purpose?

14. Figures 4A, 4B, 4C should have a common legend as an integrated paragraph. Alternatively they should be renumbered to Figures 4, 5, 6.

15. Line 397. Direct detection OF WHAT in clinical samples?

16. The bottom part of Fig. 6 presents a specific knowledge about manipulations in the course of some assay that is abundant for review paper.

17. LAMP is not the only isothermal amplification technique. Are other techniques used / may be used for the control of antibiotic resistance?

18. It would be useful to differentiate prototype developments presented in publications and the approaches already implemented to practice. A table with commercialized developments would be useful in final part of the review.

Reviewer 2 Report

Authors summarized the LFIA researches focused on the AMR detection. The topic of this manuscript is well fit to the Diagnostics and the logic of manuscript is scientifically sound. The references were also composed of the latest papers, so it is appropriate to look at recent trends. I would recommend to publish this article after modification as below:

1.     In title, acronyms, LFIA and AMR, seem not suitable. Using full word will be helpful for better understanding and authors’ interesting.

2.     For better understanding and to compare the recent technologies, creation of a table of researches referred in this manuscript with columns of assay performance such as target analyte, LOD, dynamic range, assay time, and assay principal will be helpful.

3.     Format of this manuscript should be carefully checked. (Especially, references part)  

Round 2

Reviewer 1 Report

The changes made to the manuscript reflect the main recommendations of the reviewer, and the proposed alternatives on some points are acceptable. The manuscript can be recommended for publication